# Diagnosis and Treatment of Central Serous Chorioretinopathy in Patients with Scleritis

**DOI:** 10.3390/medicina59050949

**Published:** 2023-05-15

**Authors:** Yu-Chien Tsai, Yann-Guang Chen, Yueh-Chang Lee, Yih-Shiou Hwang, Yun-Hsiu Hsieh

**Affiliations:** 1Department of Ophthalmology, Taoyuan Armed Forces General Hospital, Taoyuan 325, Taiwan; 2Department of Ophthalmology, Tri-Service General Hospital, National Defense Medical Center, Taipei 11490, Taiwan; 3Department of Ophthalmology, Hualien Tzu Chi Hospital, Buddhist Tzu Chi Medical Foundation, Hualien 970, Taiwan; 4Department of Ophthalmology, Chang Gung Memorial Hospital, Linkou Medical Center, Taoyuan 333, Taiwan; 5College of Medicine, Chang Gung University, Taoyuan 333, Taiwan; 6Department of Ophthalmology, Chang Gung Memorial Hospital, Xiamen 361000, China; 7Department of Ophthalmology, Jen-Ai Hospital Dali Branch, Taichung 41265, Taiwan

**Keywords:** scleritis, central serous chorioretinopathy, steroid, treatment

## Abstract

Central serous chorioretinopathy (CSCR) is characterized by central neurosensory retinal detachment from the retinal pigment epithelium. While the association between CSCR and steroid use is widely recognized, it is difficult to distinguish whether the subretinal fluid (SRF) in ocular inflammatory disease results from steroid use or an inflammation-related uveal effusion. We report the case of a 40-year-old man who presented to our department with intermittent redness and dull pain in both eyes that had persisted for three months. He was diagnosed with scleritis with SRF in both eyes and steroid therapy was started. Inflammation improved with steroid use, but SRF increased. This indicated that the fluid was not caused by the posterior scleritis-related uveal effusion but by steroid use. SRF and clinical symptoms subsided after steroids were discontinued completely and immunomodulatory therapy was initiated. Our study highlights that steroid-associated CSCR must be considered in the differential diagnosis of patients with scleritis, and prompt diagnosis with an immediate shift from steroids to immunomodulatory therapy can resolve SRF and clinical symptoms.

## 1. Introduction

Central serous chorioretinopathy (CSCR) is characterized by fluid accumulation under the retina that causes central neurosensory retinal detachment from the retinal pigment epithelium (RPE) over the macula. Male gender and age between 20 and 60 years are commonly associated with CSCR. The exact cause of CSCR is not fully understood, but it is thought to be multifactorial. Several risk factors have been identified, including stress, type A personality, steroid use, pregnancy, hypertension, and obstructive sleep apnea [1]. The symptoms of CSCR vary, but most patients experience a decrease in visual acuity, distorted vision, micropsia, metamorphopsia, and reduced contrast sensitivity [2]. The diagnosis of CSCR is usually made by a comprehensive eye exam, including a dilated fundus examination, optical coherence tomography (OCT), and fluorescein angiography (FA) [3,4]. Most cases of CSCR are self-limited and resolve within a few months without any intervention. However, in some cases, the fluid can persist, leading to chronic CSCR, which can result in permanent vision loss or recurrence. Treatment options for CSCR include observation, cessation of steroids, laser photocoagulation, intravitreal injections of anti-VEGF (vascular endothelial growth factor) agents, and oral or topical medications [1,2,4].

The association between CSCR and steroid use is widely recognized, but the diagnosis remains challenging in patients using steroids for ocular inflammatory disease [1]. Khairallah et al. described patients with CSCR who developed the condition following treatment with an oral, intravenous, or periocular form of corticosteroid therapy for uveitis [5]. Baumal et al. described a patient with CSCR who developed the condition following treatment with periocular injection of triamcinolone acetonide for anterior uveitis [6]. Majumder et al. reported the occurrence of CSCR in ocular inflammatory diseases, including uveitis and scleritis, which are mainly treated using steroids. However, distinguishing whether the subretinal fluid (SRF) results from CSCR or inflammation-related uveal effusion is difficult [7].

Scleritis is a rare but serious inflammatory disease that affects the white outer layer of the eye called the sclera [8]. The sclera is the tough, fibrous tissue that protects the eye and gives it its round shape. It can occur at any age, but it is most common in people between the ages of 30 and 50. Women are more likely to develop scleritis than men [9]. The symptoms of scleritis can vary depending on the severity of the condition, but they typically include severe eye pain, redness of the eye, blurred vision, and photophobia [8,9]. Scleritis can be caused by several underlying conditions, including rheumatoid arthritis, lupus, and other autoimmune disorders. It can also be caused by infections, such as herpes zoster, or by trauma to the eye [9,10,11]. It can be classified as anterior and posterior scleritis. In severe forms of posterior scleritis, inflammation may involve other ocular structures, causing retinal necrosis, optic-nerve edema, or uveal effusion which includes SRF, exudative retinal detachment, or choroidal detachment [12,13,14]. The main treatment of non-infectious scleritis is anti-inflammatory agents, including non-steroidal anti-inflammatory drugs (NSAID), steroids, or immunomodulatory therapy (IMT) which includes nonbiologic IMT agents and biologic agents [15,16]. SRF may appear in some cases of scleritis. Therefore, it is difficult to distinguish whether CSCR, occurs as a result of steroid use or is an inflammation-related uveal effusion.

Here, we report a case of scleritis with CSCR, that presented with symptoms such as red eyes and dull pain, was initially misdiagnosed as an inflammation-related uveal effusion and was treated with steroids. SRF subsided after steroids were discontinued completely.

## 2. Case Presentation

A 40-year-old man, an office administrator, denied having any systemic disease and has no known allergies, presented to our department with intermittent redness and dull pain in both eyes that had persisted for 3 months. The patient was previously diagnosed with diffuse scleritis in both eyes at another hospital, and a high dose of oral steroids was prescribed. However, the patient adjusted the steroid dose depending on the severity of his symptoms. The patient’s symptoms had never subsided completely in the past 3 months.

Our initial ocular examination revealed a best-corrected visual acuity (BCVA) of 0.2 in the right eye and 0.3 in the left eye, with a refractive error of +7 diopters hyperopia in both eyes. The intraocular pressure was 14 mmHg in the right eye and 15 mmHg in the left eye. The sclera of both eyes was diffusely reddish and edematous (Figure 1). No cells were observed in the anterior or vitreous cavities. Dilated fundus examination revealed clear vitreous in both eyes and a necrotic retinal patch over the inferotemporal arcade of the right eye (Figure 2). Macular OCT revealed subfoveal fluid in the right eye (Figure 3). FA revealed multiple leakage points in the posterior poles of both eyes (Figure 4). Indocyanine green angiography (ICGA) revealed some leaking points and multiple hypocyanescent dots giving a mottled appearance surrounded by a hypercyanescent area (Figure 4). There were no oral ulcers or skin lesions. The patient was diagnosed with diffuse scleritis in both eyes with retinal necrosis and SRF in the right eye. Our laboratory results revealed a normal range of the complete blood count (CBC), differential count, and c-reactive protein (CRP), but a mildly elevated ESR (21 mm/h) and negative results for syphilis, toxoplasmosis, tuberculosis serum test, and serum ANA, ANCA, and RF. We excluded infection etiology, so oral prednisolone 0.5 mg/kg/day and mycophenolic acid 720 mg/day was restarted.

Three weeks later, the redness and dull pain in the eyes subsided, but BCVA worsened to 0.2 in the right and 0.1 left eyes, respectively. Subsequent fundus photography revealed a large bulging elevation over the macula of the left eye. Macular optical coherence tomography revealed a massive amount of SRF in both eyes (Figure 3). Considering the improvement in clinical symptoms, SRF was suspected to be caused by long-term steroid-induced CSCR rather than scleritis-associated uveal effusion. Therefore, oral prednisolone was tapered as soon as possible with a reduction of 10 mg every three days. However, ocular redness and dull pain worsened after the steroid dosage was tapered to 20 mg/day. Therefore, we included cyclosporine 200 mg/day as the second IMT. Two months after the steroid was completely discontinued, SRF resolved completely, and symptoms, including dull ocular pain and redness, subsided (Figure 1 and Figure 3). BCVA improved to 0.5 in both eyes.

## 3. Discussion

The occurrence of CSCR in ocular inflammatory diseases is a diagnostic and therapeutic dilemma. Steroids are the mainstay treatment for ocular inflammatory disease and may be associated with CSCR; however, SRF may appear in both CSCR and uveal effusion. We report a case of scleritis with CSCR that was initially misdiagnosed as inflammation-related uveal effusion and subsided completely following the discontinuation of steroids. This is the first reported case of scleritis that initially presented with CSCR and was adequately treated.

The association between CSCR and several forms of steroid use is widely recognized [3,6,17,18,19,20,21,22]. Khairallah et al. described 20 patients with CSCR who developed the condition following treatment with corticosteroid therapy (oral prednisone in 12 patients, intravenous methylprednisolone in 1 patient, and periocular triamcinolone acetonide in 1 patient) for uveitis [5]. Baumal et al. described a patient with CSCR who developed the condition following treatment with periocular injection of triamcinolone acetonide for anterior uveitis [6]. However, the exact mechanism by which steroids contribute to the development of CSCR is not completely understood. Nevertheless, several theories have been proposed. First, steroids may cause constriction of the blood vessels in the choroid, which can lead to a backup of fluid and subsequent leakage into the retina [5,23]. Second, steroids may cause fluid buildup under the RPE, which can interfere with its ability to transport nutrients and waste products to and from the retina [24]. Third, steroids may disrupt the outer blood-retinal barrier [25]. Additionally, last, steroids can also affect the body’s hormonal balance, such as cortisol, and elevated cortisol levels have been linked to an increased risk of CSCR [5,23]. In our case, both scleritis and steroids may disrupt the blood vessel barrier, which increased the risk of SRF accumulation [1].

Scleritis may cause secondary uveal effusion, which may present as SRF or even serous retinal detachment [13]. One of the potential causes of uveal effusion is increased pressure within the eye, which can result from inflammation and thickening of the sclera in scleritis. The sclera normally acts as a barrier that helps to maintain the shape of the eye and prevent the buildup of fluid within the uvea. However, when the sclera becomes inflamed and thickened, it can lose some of its elasticity, which can lead to increased pressure within the eye and uveal effusion [9]. In addition, scleritis can also disrupt the blood vessels in the uvea, which can contribute to the accumulation of fluid [5,14]. The inflammation and increased pressure can cause the blood vessels to leak fluid into the surrounding tissue, which can lead to uveal effusion. Thus, it is challenging to determine whether the SRF is related to uveal effusion or CSCR in scleritis cases. In a case series of two cases of CSCR in patients with scleritis, the authors did not describe the cases in detail [7]. Khairallah et al. report one case of CSCR in a patient with scleritis; CSCR did not appear initially but appeared after steroid use [5]. In contrast, in our case, CSCR appeared initially, which made it more difficult to judge the etiology.

FA and ICGA are helpful for differentiating the etiology of SRF in ocular inflammatory diseases [3,19]. In CSCR, FA may reveal pinpoint leakages with or without a smoke-stack configuration, and ICGA may reveal leaking points and diffuse late choroidal hyperfluorescence [26,27]. However, in scleritis, these images may appear similar to those of CSCR, with multiple pinpoint leakages on FA and diffuse zonal hyperfluorescence, pinpoint hyperfluorescence, or a mottled pattern on ICGA. In our case, inflammation improved significantly with steroid use, but SRF increased, indicating the possibility that the fluid was not caused by the uveal effusion but by steroid usage. We believe that the changes in clinical signs in response to medication are useful to differentiate this situation.

## 4. Conclusions

It is challenging to diagnose and manage CSCR in scleritis, and the possibility of steroid-associated CSCR in patients with scleritis must be considered. Prompt diagnosis and immediate shift from steroids to IMT may stop this vicious cycle.

## Figures and Tables

**Figure 1 medicina-59-00949-f001:**
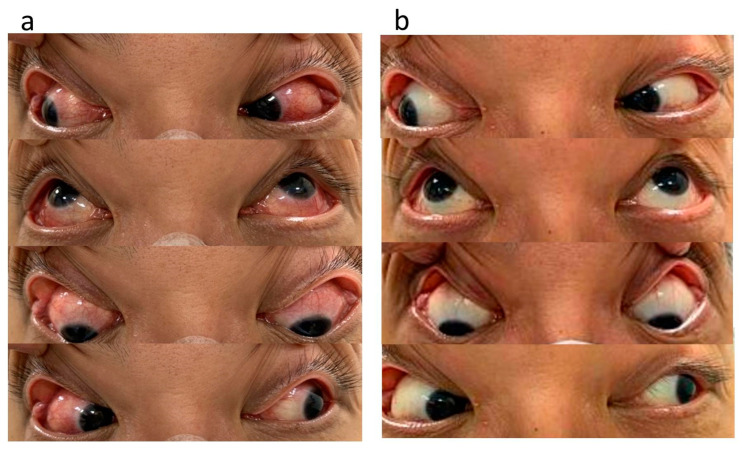
External photographs. At the initial presentation, diffuse injection and edema over the inferior and superior portion of the sclera was observed in both eyes (**a**). After adding cyclosporine as the second immunomodulatory therapy (IMT) for two months and completely discontinuing oral corticosteroid therapy, the redness of the sclera and dull ocular pain had subsided (**b**). The eyes are shown in the right, up, down, and left gazes.

**Figure 2 medicina-59-00949-f002:**
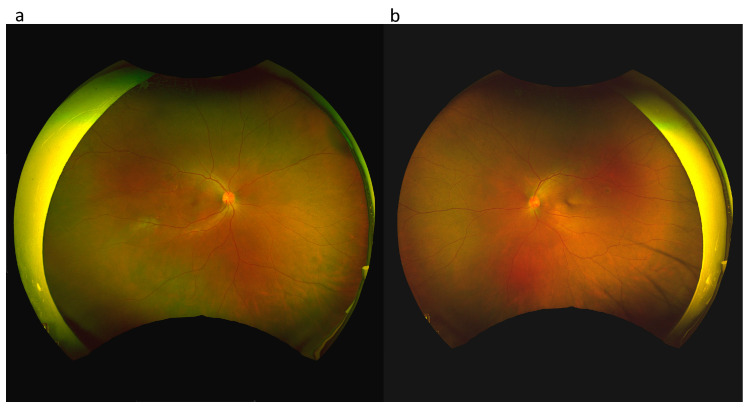
Optomap at the initial presentation of the right eye (**a**) and the left eye (**b**). A necrotic retinal patch over the inferotemporal arcade of the right eye (**a**).

**Figure 3 medicina-59-00949-f003:**
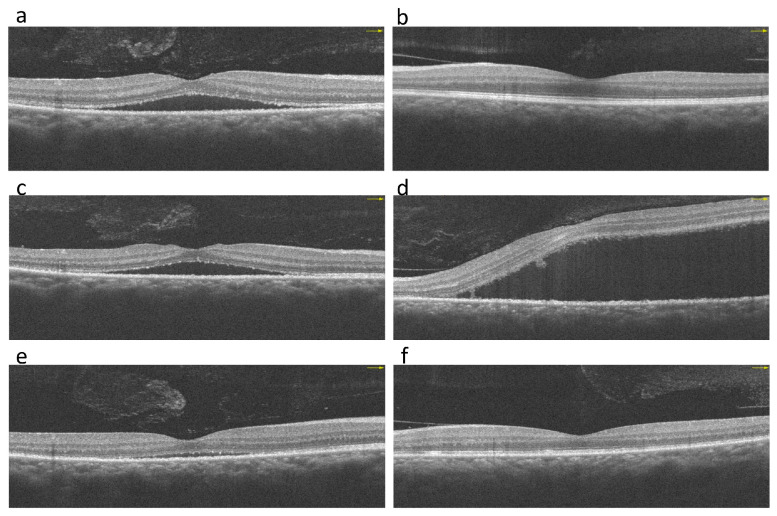
Macular optical coherence tomography (OCT) at the initial presentation (**a**,**b**). Subfoveal fluid is observed in the right eye (**a**). Macular OCT 3 weeks after oral corticosteroid treatment shows increased subfoveal fluid in the right and left eyes (**c**,**d**). Two months after the steroid was completely discontinued, subfoveal fluid has resolved in both eyes (**e**,**f**).

**Figure 4 medicina-59-00949-f004:**
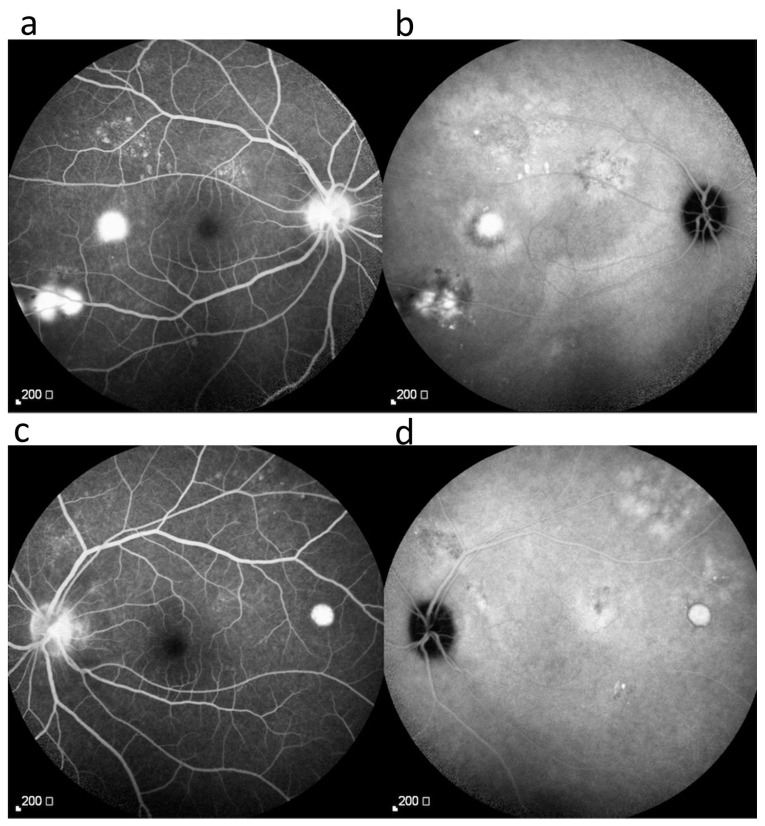
Fluorescein angiography shows optic nerve staining and multiple leaking points at the posterior pole in both eyes (**a**,**c**). Indocyanine green angiography illustrates some leaking points and multiple hypocyanescent dots giving a mottled appearance surrounded by a hypercyanescent area (**b**,**d**).

## Data Availability

All data are included in the manuscript.

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
