# Peer review of "Diagnosis and Treatment of Central Serous Chorioretinopathy in Patients with Scleritis"

_medicina, 2023, doi:10.3390/medicina59050949_

Round 1

Reviewer 1 Report (Previous Reviewer 2)

The author responses very well and the revised manuscrption is satisfing. So I agree to publish this paper on Medicina.

Author Response

Thank you for your time and effort in reviewing our manuscript. The feedback is inspiring.

Reviewer 2 Report (New Reviewer)

This case report is well documented and relevant to the audience of the Journal. I believe it underlines the need for judicious steroid use in ocular inflammation, and the risks it may associate, such as CSCR. It is important to not dismiss other therapeutical options, such as immunomodulation or non-steroidal anti inflammatory drugs in cases such as the present bilateral scleritis.

One mention I would like to add - the diagnosis (line 77) is diffuse scleritis with retinal necrosis in both eyes, while only in RE is described retinal necrosis (line 71). Please adjust accordingly.

Author Response

Thank you so much for catching these errors. The necrotic retinal patch over the inferotemporal arcade was noted only in the right eye(line 65-66). We have now corrected the diagnosis(line 72-73); however, the labeled line seems to be different.

This manuscript is a resubmission of an earlier submission. The following is a list of the peer review reports and author responses from that submission.

Round 1

Reviewer 1 Report

Central serous chorioretinopathy (CSCR) is a major cause of vision threat among middle-aged male individuals, it has been wide accepted that half-dose photodynaic therapy plays an important role in the treatment of CSCR, and steroid is not recommended. However, immunosuppressant such as cyclosporine can decrease this type of inflammation. So I don't think this manuscrption has high scientific qualility. Additionally, the necessary figures, such as B scan, fundus photography before and after cyclosporine should be provided, external photographs has no helpness to this paper.

Author Response

Point 1: Central serous chorioretinopathy (CSCR) is a major cause of vision threat among middle-aged male individuals, it has been wide accepted that half-dose photodynaic therapy plays an important role in the treatment of CSCR, and steroid is not recommended. However, immunosuppressant such as cyclosporine can decrease this type of inflammation. So I don't think this manuscrption has high scientific qualility. Additionally, the necessary figures, such as B scan, fundus photography before and after cyclosporine should be provided, external photographs has no helpness to this paper.

Response 1: The diagnosis delimma in our case is which caused the subretinal fluid (SRF), the scleritis-related inflammation or steroid-associated CSCR. Severe posterior scleritis may complicated with exudative retinal detachment, which causes large SRF. However, steroid, the main treatment of scleritis, may be related to CSCR, which also causes SRF. We judged this situation by the degree of redness of sclera (by external photographs) after steroid use, which is the main sign of slceritis. The redness subsided after steroid use, which means the inflammation became less; however, the SRF became more, which means the SRF was not related to inflammation, but steroid. Besides, thank you for your recommendation of performing B scan, we will remember in other similar cases. We did perform fundus photography and we had added the initial Optomap into the manuscript as Figure 2. Thank you for bring this to our attention.

Reviewer 2 Report

The case report carried out, “Diagnosis and treatment of central serous chorioretinopathy in patients with scleritis” is interesting because they present a clear example in which two pathologies occur, making a diagnosis and management of CSCR in scleritis, and the possibility of steroid-associated CSCR in patients with scleritis must be considered. They emphasize that rapid diagnosis and immediate treatment change can stop this vicious circle. The aims of the study are satisfied during the results, discussion and conclusion.

To improve the case report, I propose the following details to the authors:

Did the patient present any other previous systemic pathology for which he was taking medication? Did you have any known allergies?

What was his job?

These factors are important in the presentation of the case since they can give some more clue where the pathology can come from.

In patients with CSCR, a decrease in BCVA is obtained, but it is also common for these patients to report decreased contrast sensitivity and changes in color vision. Were these two aspects considered in this case or did the patient refer to any of them in his symptoms?

Author Response

Point 1: Did the patient present any other previous systemic pathology for which he was taking medication? Did you have any known allergies?

What was his job?

These factors are important in the presentation of the case since they can give some more clue where the pathology can come from.

Response 1: The patient is an office administrator and has no systemic disease and no known allergies. He was taking oral steroid previous to visiting our out patient department owing to the off-and-on symptoms, red eyes and dull pain, and was given diagnosis of scleritis. There’s also no obvious contact history.

The corrected part is on page 2 line 50 of the attached file - medicina-2205614, and as following:
“A 40-year-old man, an office administrator, denied having any systemic disease and has no known allergies, presented to our department with intermittent redness and dull pain in both eyes that had persisted for three months.”

Point 2: In patients with CSCR, a decrease in BCVA is obtained, but it is also common for these patients to report decreased contrast sensitivity and changes in color vision. Were these two aspects considered in this case or did the patient refer to any of them in his symptoms?

Response 2: The patient did not had any obvious change in color vision during the whole disease course. However, we did not perform the contrast sensitivity test in our case. Thank you for your kind reminder and we will improve in other similar situation . We had edited our manuscript.

The corrected part is on page 2 line 57 of the attached file - medicina-2205614, and as following:
“Ocular examination revealed a best-corrected visual acuity (BCVA) of 0.2 in the right eye and 0.3 in the left eye, with a refractive error of + 7 diopters hyperopia in both eyes and there was no obvious change of color vision in both eyes.”

Reviewer 3 Report

Dear Editor,

There is insufficient information to state: "In our case, inflammation improved significantly with steroid use, but SRF increased, indicating that the fluid was not caused by the inflammation itself, but by steroid usage. We believe that the changes in clinical signs in response to medication is useful to differentiate this situation." Once the SRF worsened, another angiofluoresceinography would have helped to establish the diagnosis in this case. Or the writing would be: It is possible that, in our case, SRF increased because of steroid usage...

Also, in the same paragraph:

We believe that the changes in clinical signs in response to medication is (must be "are") useful to differentiate this situation.

The manuscript must have English corrections.

Author Response

Point 1: There is insufficient information to state: "In our case, inflammation improved significantly with steroid use, but SRF increased, indicating that the fluid was not caused by the inflammation itself, but by steroid usage. We believe that the changes in clinical signs in response to medication is useful to differentiate this situation." Once the SRF worsened, another angiofluoresceinography would have helped to establish the diagnosis in this case. Or the writing would be: It is possible that, in our case, SRF increased because of steroid usage...

 Response 1: We did not perform another angiofluoreceinography when the SRF worsened. Therefore, we corrected the content as your suggestion.
The corrected part is on page 5 line 133 of the attached file - medicina-2205614, and as following:
“In our case, inflammation improved significantly with steroid use, but SRF increased, indicating the possibility that the fluid was not caused by the inflammation itself, but by steroid usage.”

Point 2: Also, in the same paragraph:

We believe that the changes in clinical signs in response to medication is (must be "are") useful to differentiate this situation.

The manuscript must have English corrections.

Response 2: Thank you for your kind reminder. We’ve corrected it. The manuscript had been edited by professional editors at Editage and has an editing certification.

The corrected part is on page 5 line 135 of the attached file - medicina-2205614, and as following:
“We believe that the changes in clinical signs in response to medication are useful to differentiate this situation.”

Round 2

Reviewer 1 Report

The author responses very well and the revised manuscrption is satisfing. So i agree to publish this paper on Medicina.